# Ketogenic Diet and Thyroid Function: A Delicate Metabolic Balancing Act

**DOI:** 10.3390/cimb47090696

**Published:** 2025-08-28

**Authors:** Petar Vranjić, Mladen Vuković, Senka Blažetić, Barbara Viljetić

**Affiliations:** 1Department of Nephrology, Internal Medicine Clinic, Osijek University Hospital, University Hospital Center Osijek, 31000 Osijek, Croatia; vranjicp@gmail.com; 2Department of Otorhinolaryngology, County General Hospital Vinkovci, 32100 Vinkovci, Croatia; mlidvu291@gmail.com; 3Department of Biology, Josip Juraj Strossmayer University of Osijek, 31000 Osijek, Croatia; senka@biologija.unios.hr; 4Department Medical Chemistry, Biochemistry and Clinical Chemistry, Faculty of Medicine Osijek, Josip Juraj Strossmayer University of Osijek, 31000 Osijek, Croatia

**Keywords:** ketogenic diet, thyroid function, inflammation, triiodothyronine

## Abstract

The ketogenic diet (KD), a high-fat, low-carbohydrate diet, causes profound metabolic adaptations that go beyond energy production and affect endocrine function and thyroid hormone regulation. By shifting the body’s primary fuel source from glucose to fatty acids and ketones, the KD alters insulin signaling, inflammation levels and deiodinase activity, which together affect thyroid hormone metabolism. While this metabolic shift offers potential benefits such as improved insulin sensitivity and reduced systemic inflammation, it also raises concerns about reduced triiodothyronine (T3) levels and altered hypothalamic–pituitary–thyroid (HPT) axis dynamics. This review explores the mechanisms by which the KD affects thyroid function, highlighting both the potential therapeutic benefits and associated risks. Special attention is given to how genetic predispositions, gut microbiota composition and sex-based hormonal differences influence thyroid adaptation to a KD. In addition, there are indications that the influence of the KD on cell metabolism could have therapeutic potential in conditions such as autoimmune thyroid diseases and thyroid cancer. Understanding the delicate balance between the benefits and risks of KD for thyroid health is essential for optimizing its clinical applications and defining individual nutritional strategies.

## 1. Introduction

The thyroid gland, a vital endocrine organ, plays a central role in metabolism, growth and overall physiological homeostasis by secreting the hormones thyroxine (T4) and triiodothyronine (T3). Anatomically, the thyroid gland consists of two lobes that are connected by a central isthmus. It is located in the anterior part of the neck, covering the trachea and positioned below the thyroid cartilage of the larynx [1]. It works in close coordination with the parathyroid glands, which regulate calcium metabolism [2]. Thyroid function is regulated by the hypothalamic–pituitary–thyroid (HPT) axis, in which the hypothalamus secretes thyrotropin-releasing hormone (TRH) to stimulate the pituitary gland to secrete thyroid-stimulating hormone (TSH). TSH in turn regulates the production and secretion of the thyroid hormones T4 and T3. Thyroid hormones influence almost every cell in the body by modulating gene expression and protein synthesis [1]. T4 acts mainly as a prohormone, while T3 is the biologically active form that directly affects metabolism, including basal metabolic rate (BMR), lipid metabolism, thermogenesis and carbohydrate metabolism [3]. In addition, thyroid hormones affect mitochondrial function, energy expenditure and thermoregulation and are therefore essential for metabolic homeostasis [4].

Considering the thyroid’s crucial role on metabolic processes, understanding its interaction with dietary interventions such as the ketogenic diet (KD) is essential, especially as the KD is increasingly used in managing body weight and metabolic disorders. The ketogenic diet is a high-fat, low-carbohydrate diet that induce ketosis, a metabolic state in which the body shifts from glucose to ketone bodies as its primary energy source [5]. Ketogenesis occurs primarily in the liver, where fatty acids are broken down to produce three main ketone bodies: beta-hydroxybutyrate (BHB), acetoacetate (AcAc), and acetone. BHB is the most abundant ketone body and serves as the primary energy substrate for tissues such as the brain, muscles, and heart, especially during low carbohydrate intake or fasting. By providing an alternative fuel source, ketone bodies enable the body to maintain energy production in the absence of glucose, significantly modulating insulin dynamics, mitochondrial function and hormonal regulation. The KD has been shown to be effective in weight loss and in the treatment of conditions such as epilepsy [6] and type 2 diabetes [7,8]. Importantly, the ketogenic diet is not a single regimen but includes several variants such as the classic ketogenic diet and the modified Atkins diet, which differ in macronutrient composition and may differ in both metabolic impact and endocrine adaptations [9,10]. However, the effects on thyroid function are still the subject of ongoing research and debate. Recent evidence suggests that the KD significantly affects thyroid hormone levels, particularly by reducing T3, TSH, TRH [11,12] while increasing T4 circulating levels. This decrease is thought to represent an adaptive response to reduced carbohydrate availability, as lower insulin and glucose levels impair the activity of deiodinase enzymes (DIO) that convert T4 to active T3 in peripheral tissues [13]. In addition, reverse T3 (rT3), an inactive form of T3 [1], may increase during ketogenic adaptation, potentially further reducing the availability of active thyroid hormone. While these adaptations may help optimize energy efficiency, the long-term effects of prolonged T3 suppression remain poorly understood, particularly in individuals with pre-existing thyroid dysfunction or those receiving thyroid hormone replacement therapy. Consequently, regular monitoring of thyroid function is recommended in individuals following KD, particularly those with conditions such as Hashimoto’s thyroiditis [12].

As the popularity of the KD continues to grow, it is important to understand its effects on thyroid health and function. The aim of this review is to evaluate the effects of the ketogenic diet on thyroid function, examining both the metabolic benefits and potential endocrine risks. Considering the thyroid gland’s central role in regulating metabolism, dietary patterns that fundamentally alter metabolic processes, such as the KD, need to be carefully studied [10]. A comprehensive understanding of these interactions is crucial for the development of evidence-based dietary recommendations that optimize thyroid health while maximizing the metabolic benefits of the KD. Future research should be focused on clarifying the specific mechanisms by which the KD affects thyroid function, long-term thyroid hormone dynamics, as well as systematically investigating outcomes of prolonged KD use in diverse populations in order to explore the potential therapeutic role of the KD in neurological and psychiatric disorders [12,14,15].

## 2. Pathways of Thyroid Regulation Under the Ketogenic Diet

The KD induces a significant metabolic shift by significantly reducing carbohydrate intake and increasing fat consumption, which leads to ketosis. In this state, the body switches from relying on glucose to fat oxidation producing ketone bodies as an alternative energy source when carbohydrates become scarce. This fundamental metabolic adjustment affects multiple hormonal systems, with the thyroid gland being especially sensitive to changes in energy availability and nutrient intake.

### 2.1. The HPT Axis and Metabolic Regulation

The HPT axis governs the regulation of thyroid hormone production through a tightly coordinated negative feedback loop. In this axis, the hypothalamus secretes TRH, which stimulates the anterior pituitary to release TSH. TSH then acts on the thyroid gland to promote the synthesis and secretion of T4 and T3, with T3 being the metabolically active form. The majority of circulating T3 is generated not directly by the thyroid, but via peripheral deiodination of T4 by iodothyronine deiodinases—primarily DIO1 in the liver and kidneys, and DIO2 in the brain, pituitary, and skeletal muscle [16,17].

Under ketogenic dietary conditions, the HPT axis undergoes characteristic adaptations in response to sustained carbohydrate restriction and altered metabolic signaling. One of the most consistent findings is a reduction in circulating free T3 (fT3) levels, often without a corresponding rise in TSH, suggesting a non-pathological downregulation of thyroid activity [10,12]. This T3 reduction is hypothesized to mimic the physiological response seen during fasting or marked caloric restriction, while TSH remains stable because central feedback perceives the lower T3 as an adaptive, not a pathological, state. The mechanistic basis for this adaptation involves reduced activity of deiodinase enzymes. Low carbohydrate availability decreases insulin secretion, which in turn suppresses deiodinase activity and impairs T3 production [12,18]. Simultaneously, levels of reverse T3 (rT3)—an inactive isomer of T3—may rise, by activation of DIO3, further reducing the pool of biologically active thyroid hormone. This shift in DIO activity appears to be a deliberate physiological response to lower energy availability, aimed at reducing basal metabolic rate and preserving energy stores [19,20].

Importantly, studies consistently report that TSH remains within the normal range during KD adherence, even as T3 declines [10,12]. This dissociation suggests that lower T3 on KD should not be automatically interpreted as hypothyroidism, especially in the absence of clinical symptoms. Rather, it reflects a homeostatic recalibration in which the hypothalamus and pituitary do not perceive the lower T3 as a pathological deficit, likely due to compensatory changes in cellular energy signaling and mitochondrial function. Additionally, ketone bodies themselves, particularly BHB, may influence HPT axis signaling. Experimental findings suggests that BHB can modulate TRH and TSH secretion at the hypothalamic and pituitary levels, possibly dampening the drive for increased thyroid hormone production under low-glucose conditions [21,22]. This interaction further supports the hypothesis that the HPT axis adapts to ketosis through multiple overlapping feedback mechanisms that prioritize metabolic efficiency over hormone abundance.

In summary, the ketogenic diet induces a characteristic alteration in HPT axis dynamics marked by reduced T3 levels, stable TSH, and potential increases in rT3. These changes likely represent adaptive, non-pathological responses to reduced carbohydrate availability, but may become clinically relevant in individuals with existing thyroid dysfunction or impaired deiodinase activity (Figure 1).

### 2.2. Hormonal Modulators of Thyroid Function

Beyond direct nutrient effects, the KD induces substantial shifts in hormonal signaling that indirectly affect thyroid hormone metabolism and HPT axis regulation. Four hormones—leptin, insulin, cortisol, and ghrelin—are particularly important in this context due to their regulatory influence on TRH and TSH secretion, deiodinase activity, and energy balance.

Leptin, a hormone produced by fat cells, is a key regulator of energy balance and directly influences the TRH expression in the hypothalamus. In energy-replete states, leptin stimulates TRH production, supporting thyroid hormone synthesis and maintaining BMR [23,24]. Under KD, weight loss and reduced adiposity lead to significant decreases in leptin levels, which decrease TRH output and downregulate TSH secretion [25,26]. This cascade contributes to reduced thyroid hormone production and is part of the body’s adaptive effort to conserve energy under conditions of reduced caloric and carbohydrate intake. In contrast, studies in children and adults with GLUT1-deficiency syndrome and drug-resistant epilepsy on long-term KD found little to no change in leptin [15]. Despite the lower leptin levels, during KD, leptin sensitivity in the brain increases [27]. This indicates an additional adaptive response by the body that becomes more responsive to leptin’s signals, promoting satiety and improved appetite control, even with less leptin present.

Insulin is a key metabolic hormone that can regulate DIO2 activity, particularly in brown adipose tissue, skeletal muscle and glial cells thereby facilitating the local conversion of T4 to active T3. This regulation is especially important in tissues like skeletal muscle, brown adipose tissue, and glial cells, where local T3 production supports energy metabolism. Insulin upregulates DIO2 activity by relieving transcriptional repression, thereby enhancing local T3 generation in response to nutrient availability [18,28]. In the context of KD, insulin levels fall sharply due to low carbohydrate intake, thereby reducing deiodinase stimulation and contributing to lower circulating T3 concentrations [11]. This may be beneficial in individuals with insulin resistance or type 2 diabetes, where improved insulin sensitivity under KD supports metabolic rebalancing. However, for individuals with thyroid dysfunction, diminished insulin signaling may further impair T3 production and exacerbate hypothyroid symptoms [29,30,31].

Cortisol, a glucocorticoid released in response to stress, typically rises during the early phase of ketogenic diet adaptation, reflecting a heightened stress response that mobilizes gluconeogenic substrates and helps maintain blood glucose. In both human and animal studies, this short-term elevation peaks within the first two days of carbohydrate restriction and may remain above levels seen on high-carbohydrate diets for several weeks [32,33]. However, with continued adherence—around 6–8 weeks—cortisol often normalizes or even falls below baseline, as demonstrated in obese individuals on very low-calorie ketogenic regimens, suggesting the body gradually adapts and the initial stress diminishes [34]. Importantly, sustained cortisol elevation can suppress the HPT axis by inhibiting TRH expression, reducing TSH secretion, and impairing deiodinase activity, which not only blunts thyroid hormone production but may also raise rT3 levels and exacerbate functional hypothyroidism in susceptible individuals. This dynamic underscores the need for monitoring and stress-management strategies during both the early and prolonged phases of KD adherence.

Ghrelin, primarily known as an orexigenic hormone that increases appetite, also interacts with the HPT axis, generally suppressing TSH and influencing thyroid hormone levels, but the direction and magnitude of these effects depend on physiological context and species. The relationship is complex and may serve as a regulatory mechanism linking energy status, appetite, and thyroid function [35,36]. Some evidence suggests that ghrelin levels may rise during energy restriction or fasting-like states (such as KD), potentially providing a counter-regulatory signal to maintain thyroid activity [37]. However, this effect appears to be relatively minor compared to the suppressive influence of leptin and cortisol.

Together, these shifts create a coherent endocrine profile favoring energy conservation, characterized by reduced thyroid hormone activation despite stable or only mildly suppressed central HPT signaling. While adaptive in metabolically healthy individuals, these hormonal changes may require clinical attention in patients with pre-existing thyroid disorders or stress-related endocrine dysregulation. In addition, other hormones such as glucagon, adiponectin, and growth hormone also contribute to thyroid regulation, although their role in KD is less well defined.

### 2.3. Macronutrient Composition and Thyroid Hormone Conversion

The macronutrient distribution characteristic of the ketogenic diet—very low carbohydrate, high fat, and moderate protein—plays a central role in modulating thyroid hormone dynamics, particularly the peripheral conversion of T4 to T3. Among macronutrients, carbohydrate restriction exerts the most profound effect on thyroid function due to its regulatory influence on deiodinase enzymes. Under normal dietary conditions, glucose availability and insulin signaling maintain robust deiodinase activity. However, under a KD, the sharp decline in carbohydrate intake leads to hypoinsulinemia and low glucose availability, both of which suppress deiodinase expression and activity [12,18]. This suppression results in decreased T3 production and a compensatory rise in rT3, an inactive thyroid hormone metabolite that competes with T3 for receptor binding. The resulting hormone profile mirrors that seen during fasting or severe caloric restriction, reflecting a deliberate metabolic downshift to preserve energy under conditions of limited glucose availability [20,38].

Protein intake also modulates thyroid hormone metabolism. While the ketogenic diet allows for moderate protein consumption, excessively low protein intake can become problematic. Proteins supply essential amino acids such as tyrosine, a precursor for T3 and T4 synthesis. Inadequate protein may compromise thyroid hormone biosynthesis and reduce substrate availability for key enzymes [39]. Conversely, excessive protein consumption may stimulate gluconeogenesis, potentially impeding ketosis and altering metabolic signaling. Thus, protein must be carefully balanced on the KD to preserve both ketogenesis and endocrine homeostasis.

Furthermore, the quality of fats consumed may indirectly impact thyroid hormone metabolism by modulating inflammation, oxidative stress, and insulin sensitivity. Diets high in saturated fats are linked to increased systemic inflammation and unfavorable changes in lipid profiles, both of which may indirectly impair thyroid function [40,41]. In contrast, mono- and polyunsaturated fats—particularly omega-3 fatty acids—may support anti-inflammatory pathways and preserve deiodinase activity.

Aside from general macronutrient restriction, it is important to note that the KD is not a one-size-fits-all approach. There are several variations, including the classic KD (typically a 4:1 ratio of fat to carbohydrate plus protein), the modified Atkins diet (with a more liberal protein intake and a less strict fat ratio), the medium-chain triglyceride diet (enriched with octanoic (C8) and decanoic (C10) fatty acids, which yield more ketones per kilocalorie than long-chain triglycerides), and the low glycemic index treatment (which allows a higher carbohydrate intake provided the sources have a glycemic index < 50) [9]. These differences in macronutrient composition may have important consequences for thyroid hormone regulation and endocrine adaptation. For example, a randomized, isocaloric crossover study in healthy individuals showed that a three-week sustained classic KD resulted in a significant decrease in fT3 and an increase in free T4 (fT4), indicating a shift in thyroid function despite an unchanged resting metabolic rate (RMR) [12]. In adult patients with drug-resistant epilepsy, a 12-week modified Atkins diet resulted in a similar reduction in T3 and fT3 with a concomitant increase in fT4, suggesting modest but clinically relevant thyroid suppression [42].

In sum, the macronutrient composition of the KD—especially its extreme carbohydrate restriction—plays a critical role in suppressing T3 levels through reduced deiodinase activity. A balanced protein intake is required to maintain thyroid hormone synthesis without compromising ketosis, and differences in fat quality can further influence metabolic and endocrine responses.

## 3. Immune, Inflammatory and Oxidative Modulation Under Ketogenic Diet

The KD not only alters metabolic and hormonal signaling but also modulates immune activity, systemic inflammation, and oxidative stress, all factors important for thyroid homeostasis and the pathogenesis and progression of autoimmune thyroid disease such as Hashimoto’s thyroiditis and Graves’ disease [43]. Importantly, these effects may help explain the observed changes in thyroid function and the potential risk or modulation of autoimmunity during ketogenic interventions.

### 3.1. Oxidative Stress and Mitochondrial Function

The thyroid gland is one of the most metabolically active organs in the body and has a high intrinsic demand for mitochondrial energy production. This metabolic intensity, combined with its dependence on oxidative reactions for thyroid hormone synthesis, makes the gland extremely sensitive to oxidative stress. Excessive production of reactive oxygen species (ROS) plays a well-documented role in the initiation and progression of autoimmune thyroid diseases, contributing to cellular damage, antigen presentation, and immune activation [44].

The KD is increasingly recognized for its ability to improve mitochondrial metabolism and reduce oxidative stress, both key contributors to inflammation and immune activation, including in the thyroid. By shifting energy production from glycolysis to fatty acid oxidation, KD leads to increased generation of ketone bodies such BHB and AcAC, which serve not only as efficient energy substrates but also as modulators of mitochondrial function and redox homeostasis [45,46]. In addition to lowering ROS production, the KD enhances antioxidant defense systems. BHB has been shown to activate the nuclear factor erythroid 2–related factor 2 (Nrf2), a transcription factor that induces the expression of numerous antioxidant enzymes, including superoxide dismutase (SOD) and glutathione peroxidase [47]. This mechanism provides an endogenous antioxidant shield against oxidative injury, potentially mitigating thyrocyte damage in conditions such as Hashimoto’s thyroiditis.

Furthermore, improvements in mitochondrial efficiency under the KD may enhance ATP production per unit of substrate, reducing the metabolic stress on thyroid epithelial cells [45,48]. This is particularly important in autoimmune disease states where mitochondrial dysfunction is often present. By promoting oxidative phosphorylation efficiency and lowering redox imbalance, the KD may help stabilize thyroid tissue against chronic inflammatory insults.

Taken together, these effects suggest that the KD supports thyroid health by reinforcing mitochondrial resilience and redox balance. These pathways are especially relevant in thyroid autoimmune pathogenesis, where oxidative stress serves as both a trigger and an amplifier of immune-mediated tissue destruction (Figure 2).

### 3.2. Anti-Inflammatory Effects of Ketogenic Adaptation

Chronic low-grade inflammation is increasingly recognized as a central factor in the development and progression of thyroid dysfunction, particularly in autoimmune thyroid diseases such as Hashimoto’s thyroiditis [49,50,51]. A key pathogenic mechanism involves activation of inflammasomes—especially the NOD-like receptor protein 3 (NLRP3)—which triggers the release of pro-inflammatory cytokines (interleukin-1β (IL-1β), interleukin-18 (IL-18), tumor necrosis factor-alpha (TNF-α) and interferon-gamma (IFN-γ). These cytokines promote immune cell infiltration, loss of immune tolerance and contribute to the gradual destruction of thyroid tissue.

The KD has gained attention for its systemic anti-inflammatory effects that may influence thyroid hormone regulation and immune tolerance. A key mechanism involves the action of BHB, the principal circulating ketone body during ketosis, which serves not only as an alternative energy source but also as a signaling molecule with anti-inflammatory properties. Notably, BHB inhibits the activation of the NLRP3 inflammasome, thereby reducing IL-1β and IL-18 production [12,52,53]. Through this pathway, the KD may suppress systemic inflammatory responses implicated in autoimmune thyroid pathophysiology.

In addition to BHB’s direct effects, KD appears to activate peroxisome proliferator-activated receptor gamma (PPARγ), a nuclear receptor involved in lipid metabolism and inflammation resolution. For example, in murine models of ketosis, KD-induced PPARγ activation in the brain was accompanied by decreased TNF-α, Nuclear factor kappa B (NF-κB) and cyclooxygenase-2 (COX-2) expression in hippocampal tissue [54]. Furthermore, a recent meta-analysis confirmed that ketogenic diets consistently reduce circulating TNF-α and IL-6 levels in humans [55]. Both cytokines are elevated in autoimmune thyroid disease and have been shown to impair thyroid hormone signaling and promote lymphocytic infiltration of the gland.

Collectively, these mechanisms suggest that the ketogenic diet may attenuate the inflammatory and oxidative burden on the thyroid gland by targeting key immunometabolic pathways implicated in autoimmune thyroiditis. However, while preclinical and mechanistic studies offer promising insights, the clinical relevance of these effects, particularly in terms of long-term outcomes, patient heterogeneity, and safety, remains to be clarified through well-designed human trials.

### 3.3. Autoimmune Modulation and Regulatory T Cells

Autoimmune thyroid diseases, such as Hashimoto’s thyroiditis and Graves’ disease, are marked by the breakdown of immune tolerance that leads to the chronic infiltration of autoreactive T and B lymphocytes into the thyroid gland. This immune dysregulation results in the production of autoantibodies, destruction of thyroid tissue, and progressive hypothyroidism. One of the central failures in this process is impaired regulatory T cell (Treg) function, which normally acts to suppress autoreactive immune responses and maintain self-tolerance [56,57].

Emerging evidence suggests that the KD, through the activity of its key ketone body BHB, may positively influence Treg populations and overall immune balance. BHB has been shown to modulate the metabolism of immune cells, particularly T cells, by inhibiting histone deacetylases (HDACs) and altering gene expression involved in immune regulation. This epigenetic modulation may favor the differentiation and stability of FoxP3+ regulatory T cells, which are essential for suppressing autoimmune responses and curbing inflammatory damage to the thyroid gland [58,59,60]. In animal models, BHB has been observed to increase Treg activity and suppress pro-inflammatory T helper 17 cell (Th17) responses, both of which are relevant in the pathogenesis of Hashimoto’s thyroiditis. Th17 cells are known to promote tissue inflammation and have been implicated in thyroid autoimmunity through the production of IL-17 and related cytokines. By shifting the balance from Th17 dominance toward Treg activity, the KD may help reduce the autoimmune burden and slow thyroid tissue destruction.

Furthermore, the ketogenic state may influence antigen presentation by modulating the expression of key immune-related genes and surface molecules. Ketogenic diets and ketone bodies have been shown to enhance the expression of major histocompatibility complex (MHC) class I genes and promote type I interferon signaling, thereby supporting antigen presentation and T cell priming—especially in the context of cancer immunotherapy [61,62]. Additionally, the ketogenic state activates AMP-activated protein kinase (AMPK), which further upregulates antigen presentation pathways and reduces the expression of immune checkpoint proteins such as programmed cell death 1 ligand 1 (PD-L1), collectively facilitating a more robust T cell response. While these effects have been primarily studied in tumor models, they align with the broader anti-inflammatory and immunomodulatory profile of KD observed across various chronic inflammatory conditions.

Importantly, these adaptive immune effects may not manifest uniformly across individuals. Genetic predispositions, disease stage, and baseline immune status likely determine whether KD-induced Treg enhancement and immune modulation translate into measurable clinical benefit in autoimmune thyroid disease.

## 4. Genetic and Individual Factors Influencing Response to KD

### 4.1. Genetic Determinants

Interindividual variability in thyroid hormone response to KD is partly shaped by genetic polymorphisms that modulate hormone conversion, fat metabolism, and ketone production. Several well-characterized single nucleotide polymorphisms (SNPs) have been implicated in determining how effectively individuals adapt to KD, both metabolically and hormonally.

*DIO2* (Thr92Ala) and T3 Conversion Efficiency. The DIO2 enzyme plays a critical role in converting inactive T4 into biologically active T3 within peripheral tissues [13]. A common polymorphism, Thr92Ala (rs225014), results in reduced DIO2 enzymatic activity and has been associated with blunted intracellular T3 levels, despite normal circulating thyroid hormone concentrations [63]. Under the KD where insulin and leptin levels are suppressed and deiodinase stimulation is reduced, individuals with the Ala/Ala or Thr/Ala genotype may be particularly vulnerable to functional intracellular hypothyroidism, even when serum TSH and T4 appear normal [64,65]. These individuals may experience reduced cognitive clarity, fatigue, or cold intolerance during prolonged ketosis, highlighting the importance of genotype-guided diet personalization.

*APOA2*(*−*265T > C) and Fat Metabolism. The *APOA2* gene encodes apolipoprotein A-II, a regulator of lipoprotein metabolism and satiety signaling [66]. The −265T > C variant (rs5082) influences dietary fat response: C allele carriers (especially CC homozygotes) are more prone to increased BMI, visceral adiposity, and insulin resistance when consuming diets high in saturated fat [67,68]. Given that the KD is typically high in dietary fat, individuals with the CC genotype may exhibit a less favorable lipid and insulin response, which could interfere with optimal thyroid signaling, particularly through insulin–deiodinase pathways. In such cases, a modified KD emphasizing unsaturated fats or a more moderate-fat approach may be advisable.

*PPARA* (Leu162Val) and Ketogenesis Efficiency. The peroxisome proliferator-activated receptor alpha (PPARα) is a nuclear receptor that regulates genes involved in fatty acid oxidation and ketone production [69,70]. The Leu162Val polymorphism (rs1800206) affects *PPARα* transcriptional activity: Val allele carriers may demonstrate enhanced fatty acid utilization and ketone generation, while Leu/Leu individuals may experience a more sluggish metabolic shift into ketosis [71,72]. This polymorphism could influence the rate and extent of keto-adaptation, thereby affecting the hormonal and metabolic milieu, including thyroid hormone dynamics. Poor ketogenesis efficiency may lead to prolonged stress responses (e.g., elevated cortisol), which can suppress TSH and promote reverse T3 (rT3) accumulation.

These and other genetic polymorphisms influence how individuals respond to ketogenic diets in terms of thyroid hormone activation, metabolic efficiency, and fat handling. Personalized dietary strategies based on genotype may help mitigate unwanted hormonal effects. As personalized nutrition becomes more integrated into clinical practice, genotyping for such variants may support safer and more effective long-term KD use in populations with thyroid vulnerability.

### 4.2. Sex-Based Differences and Reproductive Endocrinology

Sex-based hormonal differences significantly influence both thyroid physiology and metabolic adaptation to the KD. Women are disproportionately affected by thyroid disorders, particularly autoimmune thyroiditis, and may respond differently to nutritional interventions due to cyclical hormonal changes and differences in leptin and thermogenic signaling. Thereby, the KD tends to be more effective for weight loss and fat reduction in men compared to women [73].

Leptin, an adipose-derived hormone, is a key metabolic signal that links energy stores to both thyroid regulation and reproductive function. Leptin acts as a permissive factor for the hypothalamic–pituitary–gonadal (HPG) axis, ensuring that reproduction only proceeds when energy reserves are sufficient [74,75]. Women, who naturally exhibit higher leptin levels than men, may be more susceptible to leptin-mediated reproductive disruptions when following energy-restrictive or carbohydrate-restricted diets [26,76]. KD-induced reductions in leptin can suppress hypothalamic GnRH release, thereby decreasing FSH and LH secretion, which may lead to anovulation, shortened luteal phases, or amenorrhea—especially in lean or physically active women [77].

Estrogen plays a central role in modulating thyroid hormone dynamics, particularly through its effect on thyroxine-binding globulin (TBG). Estrogen increases hepatic synthesis of TBG, which in turn raises total T4 and T3 levels while reducing the free fraction of these hormones [78,79,80]. In premenopausal women or those on hormonal contraceptives, this shift may lead to misleading laboratory readings such as normal total T4 with low fT4 and symptoms of hypothyroidism. Under KDs, reductions in insulin and caloric intake may further decrease hepatic protein synthesis, potentially compounding this hormonal imbalance. Women with already elevated TBG levels may be particularly sensitive to this shift and may require adjustments in thyroid hormone dosing or diet composition [81,82].

Progesterone, which rises during the luteal phase of the menstrual cycle, is well-established to increase core body temperature and RMR and potentially thyroid hormone sensitivity [83]. The KD may interfere with these cyclic metabolic shifts by altering energy sensing and hormonal signaling. In women with disrupted progesterone production (e.g., anovulatory cycles), the expected luteal-phase rise in thermogenesis is diminished. This may influence the interpretation of thyroid status and basal body temperature tracking, which some practitioners use as a surrogate marker of metabolic health. Additionally, reduced T3 output on the KD may amplify luteal phase fatigue or cold sensitivity in susceptible women [84,85].

In men, testosterone promotes lean mass retention, mitochondrial function, and lipid oxidation, all of which synergize with the KD’s metabolic profile. The KD has been associated with either stable or increased testosterone levels in men, depending on total energy intake and fat composition [86,87]. Higher testosterone may enhance thyroid hormone responsiveness by improving deiodinase activity and increasing tissue sensitivity to T3 [88]. As a result, men may demonstrate greater metabolic efficiency and fewer hypothyroid-like symptoms during keto-adaptation, although this benefit may be reduced in older men with hypogonadism or androgen deficiency.

In addition to hormonal mechanisms, human data suggest sex-specific responses to ketogenic therapy for weight loss, with men often showing greater reductions in body mass and inflammation levels than women [73,89]. These differences likely reflect not only hormonal influences, but also sex-specific patterns of fat distribution, substrate utilization, and adaptive thermogenesis. However, direct evidence for sex-specific changes in thyroid hormone (T3, T4, TSH) under the KD, is limited. Taken together, these physiological differences emphasize the need for sex-specific considerations when applying KDs, particularly in individuals with thyroid disorders or reproductive problems.

## 5. Gut Microbiota, Nutrient Absorption, and SCFA Deficiency

Diet-induced alterations to the gut microbiota, nutrient bioavailability, and microbial metabolites can influence immune signaling, thyroid hormone activation, and hepatic metabolism. The KD, while beneficial in reducing inflammation, can disrupt gut microbial balance in ways that may have unintended consequences for thyroid function.

Dysbiosis—an imbalance of commensal and pathogenic microbes—is increasingly recognized as a trigger for autoimmune disease, including thyroid autoimmunity. Disrupted gut microbiota may lead to increased intestinal permeability (“leaky gut”), molecular mimicry (e.g., cross-reactivity between bacterial antigens and TPO), and activation of Th17/IL-17 pathways. These immune disruptions can initiate or exacerbate thyroid-directed autoimmunity [90,91]. A restrictive KD that diminishes microbial resilience may, paradoxically, worsen autoimmune progression in vulnerable populations—particularly without targeted prebiotic or probiotic support [92].

Standard KD protocols often eliminate or sharply reduce fiber-rich plant foods, leading to reduced intake of prebiotic fibers such as inulin, arabinoxylans, and resistant starch. This fiber restriction significantly affects the composition and diversity of the gut microbiota, particularly by depleting beneficial butyrate-producing bacteria in the *Firmicutes* phylum (e.g., *Faecalibacterium prausnitzii*, *Roseburia* spp.) [92,93]. Loss of microbial diversity may impair the gut barrier and promote metabolic endotoxemia, which in turn can trigger systemic inflammation and autoimmune activity in the thyroid gland. Reduced microbial diversity has been observed in patients with autoimmune thyroiditis, suggesting that a fiber-deficient KD may worsen dysbiosis in genetically predisposed individuals [94].

Short-chain fatty acids (SCFAs), such as butyrate, acetate, and propionate, are microbial metabolites that influence host immunity, intestinal integrity, and thyroid hormone metabolism [95]. Butyrate has been shown to upregulate DIO2 expression in enterocytes and glial cells, supporting the conversion of T4 to active T3. SCFAs also activate G-protein-coupled receptors (GPCRs) and modulate histone acetylation, thereby influencing immune tolerance [96]. A KD-induced reduction in SCFA production may therefore impair peripheral T3 activation, particularly in individuals with low baseline DIO2 activity (e.g., *Thr92Ala* polymorphism) [97,98]. This can exacerbate the risk of low intracellular T3 despite normal serum levels—a phenomenon frequently reported in hypothyroid patients during prolonged ketosis. Gut microbial health is also critical for the absorption of thyroid-relevant micronutrients, particularly iodine (needed for T4/T3 synthesis), selenium (required for deiodinase and glutathione peroxidase) and zinc (involved in TSH production and receptor binding). KD-associated dysbiosis may reduce the bioavailability of these trace elements, either through changes in pH, gut transit time, or mucosal integrity. Inadequate intake of plant-based mineral sources due to a KD further compounds this risk [94]. Iodine is essential for thyroid hormone synthesis; however, both deficiency and excess can provoke autoimmune responses. High iodine levels have been shown to increase thyroid peroxidase antibody (TPOAb) titers and may accelerate autoimmune thyroid destruction by enhancing antigen presentation and oxidative stress within the thyroid gland. Conversely, inadequate iodine, possible on strict KD protocols lacking iodized salt or seafood, can impair thyroid hormone production and exacerbate hypothyroid symptoms [99,100,101]. While butyrate-producing bacteria such as *Faecalibacterium prausnitzii* and *Roseburia* spp. are crucial for the maintenance of intestinal integrity and thereby indirectly support micronutrient absorption, the specific taxa that directly facilitate iodine, selenium, or zinc absorption in humans are still poorly characterized. Preliminary evidence suggests that probiotic supplementation, particularly with *Lactobacillus* and *Bifidobacterium* strains, may lower thyroid autoantibody titers and support micronutrient absorption [92,93], but robust data in the context of KDs are currently lacking.

The KD is notably restrictive and may result in insufficient intake of essential vitamins and minerals, including selenium, zinc, and magnesium, which are critical for thyroid function [102]. Selenium is essential for deiodinase enzymes and glutathione peroxidase, both of which protect thyroid tissue from oxidative damage [103]. Zinc is necessary for TSH production and thyroid receptor function [104]. Inadequate intake or reduced bioavailability of these minerals that are potentially influenced by dietary restrictions and changes in gut microbiota, may compromise the anti-inflammatory benefits of KD and contribute to thyroid hormone imbalance [105]. Over time, micronutrient depletion may reduce antioxidant defense and impair hormone synthesis and conversion which are key processes in thyroid function. While the KD has demonstrated potential in downregulating inflammation and enhancing immune regulation, its prolonged use may introduce risks that counteract its benefits in thyroid autoimmunity. These considerations underscore the importance of personalized implementation and regular biochemical monitoring when applying KD in patients with autoimmune thyroid disease.

A KD significantly alters bile acid metabolism by increasing lipid intake and modifying enterohepatic circulation. Secondary bile acids, which are microbially derived, regulate liver enzymes involved in T4–T3 conversion and hepatic deiodinase activity. Dysbiosis can change bile acid composition, which in turn impacts hepatic thyroid hormone metabolism and feedback signaling to the HPT axis [12,106,107]. Additionally, altered bile flow and gut–liver communication may affect thyroid hormone clearance, raising the possibility of subclinical imbalances in free hormone levels—even in euthyroid individuals on a KD.

The KD may be most effective when microbiota integrity is preserved through targeted fiber reintroduction, probiotic support, or cyclical carbohydrate inclusion to sustain SCFA levels and mineral uptake. These adjustments may protect against hidden thyroid vulnerabilities during long-term ketogenic interventions.

## 6. Clinical Impact and Population-Specific Adaptation of the Ketogenic Diet

### 6.1. Hypothyroidism and LT4 Dependence

Individuals with hypothyroidism, particularly those on levothyroxine (LT4) monotherapy, represent a sensitive population when adopting a KD [82,108], due to its impact on thyroid hormone metabolism, peripheral conversion, and diagnostic interpretation. This poses a significant concern for LT4-treated patients who rely entirely on peripheral conversion for adequate T3 supply. When conversion is impaired, patients may experience low tissue-level T3 despite normal serum TSH and T4 values. Furthermore, the metabolic stress associated with the KD, such as caloric restriction, weight loss, or increased cortisol, can increase rT3 levels, which antagonize T3 at its receptor and blunt metabolic activity. These conditions give rise to the so-called “low T3 syndrome” or “euthyroid sick syndrome,” in which patients exhibit clinical signs of hypothyroidism despite biochemical euthyroidism [109,110].

This mismatch is further complicated by the fact that standard thyroid panels typically exclude fT3 and rT3 measurements, which can render tissue-level hypothyroidism undetectable through conventional testing. In such contexts, assessing fT3 levels may provide a more accurate picture of thyroid hormone activity, particularly in individuals with genetic polymorphisms such as *DIO2 Thr92Ala*. These patients may benefit from combination therapy with liothyronine (T3) alongside levothyroxine (T4), although more targeted research is needed to confirm individualized treatment responses [111,112,113]. This strategy aims to bypass impaired conversion and provide direct hormonal activation, though it must be personalized and initiated with caution.

Another clinical challenge is the TSH–T3 mismatch: while TSH reflects central (pituitary) regulation, it does not necessarily correspond to peripheral thyroid hormone activity, especially in the altered hormonal environment induced by ketosis [114]. KD-associated changes in leptin, SCFAs, and bile acid metabolism may further uncouple pituitary feedback from tissue-level thyroid function. Clinicians are therefore encouraged to adopt a broader diagnostic approach—one that integrates clinical symptoms with extended thyroid panels, including fT3 and rT3, while also considering dietary context [115]. Patients reporting fatigue, cold intolerance, or cognitive slowing while on KD—despite normal TSH—should be assessed for potential functional hypothyroidism and evaluated for nutritional adequacy, particularly regarding selenium, iodine, and zinc status, all of which are essential for optimal thyroid hormone synthesis and conversion [116].

### 6.2. Euthyroid Adaptation and Subclinical Risk

While LT4-treated hypothyroid patients are especially sensitive to these changes, even euthyroid individuals may experience functional adaptation to a KD that mirrors elements of ‘low T3 syndrome‘. Among euthyroid individuals—those with biochemically normal thyroid function—the KD is generally well tolerated, but subtle hormonal shifts may occur, particularly in those with underlying vulnerabilities or borderline thyroid reserve. The KD has been consistently associated with modest reductions in fT3 levels without significant changes in TSH or fT4 [11,12,82]. These reductions may reflect a physiological adaptation to lower glucose availability and energy expenditure, rather than overt dysfunction, as the KD can enhance mitochondrial efficiency and reduce the need for high T3-driven metabolic output [48,117] (Figure 3). However, in individuals with low baseline T3 or marginal deiodinase activity—such as carriers of the *DIO2 Thr92Ala* polymorphism—this adaptation may lead to fatigue, cold intolerance, or cognitive fog despite normal lab values. Similarly, those transitioning from high-carbohydrate diets may experience transient suppression of T3 and elevation of rT3, particularly during early ketosis or under caloric deficit. In cases of subclinical hypothyroidism, characterized by mildly elevated TSH with normal fT4, KD may either stabilize or destabilize thyroid function depending on the individual’s energy intake, stress levels, and micronutrient status. Some reports suggest that inflammation reduction and weight loss under a KD can improve TSH levels and reduce autoimmune activity, potentially reversing subclinical hypothyroidism in metabolically inflamed individuals [12,118]. However, others may progress to overt hypothyroidism if thyroidal reserve is limited and metabolic demands are high. For these populations, careful clinical monitoring—including free T3, rT3, and thyroid antibody panels—is recommended during the initial months of KD adaptation. Dietary adequacy in iodine, selenium, and zinc remains essential, and therapeutic carbohydrate targeted intake or micronutrient support may help mitigate the drop in active thyroid hormone. Ultimately, while a KD may be metabolically advantageous for many euthyroid individuals, those with subtle thyroid dysfunction or autoimmune diathesis require proactive management to prevent hormonal imbalances from emerging during prolonged carbohydrate restriction.

### 6.3. Hashimoto’s Thyroiditis

For patients with Hashimoto’s thyroiditis, the KD offers a nuanced therapeutic potential that requires careful personalization. On the beneficial side, the KD has been shown to reduce systemic inflammation by lowering pro-inflammatory cytokines. These changes may help dampen the autoimmune response and reduce TPOAb titers in susceptible individuals [101]. However, these benefits are counterbalanced by potential risks. The diet’s induction of cortisol, especially under caloric restriction or rapid fat loss, can inhibit deiodinase activity and increase rT3 production, contributing to reduced tissue-level thyroid hormone activity [119]. Fluctuations in iodine intake are also concerning: The KD often eliminates iodized table salt and dairy, key iodine sources, while inconsistently including seaweed or fish, raising the risk of both deficiency and excess—both of which are known to trigger or worsen autoimmune thyroiditis [100]. Additionally, deficiencies in selenium and zinc, commonly observed in restrictive diets, can impair antioxidant defense (via glutathione peroxidase) and reduce deiodinase function, further diminishing thyroid hormone activation [105]. These risks highlight the importance of individualized monitoring strategies, including periodic assessment of fT3, rT3, TPOAb, selenium, iodine, and clinical symptomatology. A modified KD that includes unsaturated fats, prebiotic fiber, and micronutrient repletion may preserve immunometabolic benefits while reducing the risk of thyroid dysfunction. Ultimately, a KD may offer therapeutic benefit in Hashimoto’s when applied with attention to genetic background, disease activity, and endocrine feedback loops.

### 6.4. Hyperthyroidism and Graves’ Disease

Although much of the concern surrounding KD and thyroid function centers on hypothyroid suppression, their potential implications for hyperthyroid states—particularly Graves’ disease—deserve critical attention. Graves’ disease is characterized by immune-mediated stimulation of the TSH receptor, resulting in excessive T3 and T4 production, elevated metabolic rate, and increased oxidative stress [120,121]. A this moment, the role of the KD in Graves’ disease remains largely theoretical and underexplored.

On the potentially beneficial side, a KD-induced anti-inflammatory profile may modulate autoimmune activity in Graves’ disease, potentially reducing the inflammatory burden on the thyroid gland. Additionally, the mitochondrial efficiency and redox-enhancing effects of BHB may attenuate the oxidative stress and tissue damage often seen in hyperthyroid states [45]. A KD’s glucose-lowering effects could also theoretically stabilize adrenergic symptoms by reducing systemic excitability, although this remains speculative. However, several concerns warrant caution. In active hyperthyroidism, the body is already in a catabolic state, with increased protein turnover, hepatic gluconeogenesis, and sympathetic activation. Initiating a KD under these conditions may exacerbate energy deficits, promote muscle wasting, or induce adrenal stress. Moreover, rapid changes in carbohydrate intake may influence thyroid hormone metabolism via altered leptin and insulin signaling, possibly affecting TRH and TSH feedback loops. Patients with Graves’ ophthalmopathy may also be vulnerable to electrolyte shifts and micronutrient deficiencies (e.g., selenium, magnesium) during the early ketogenic transition.

To date, no clinical trials have assessed the KD in patients with active hyperthyroidism or Graves’ disease and therefore its utility must be considered highly speculative. Thus, any application of this dietary strategy in this population must be highly individualized and coordinated with pharmacologic management (e.g., methimazole or beta-blockers). Future research should explore whether the anti-inflammatory and mitochondrial mechanisms observed in the KD might serve as adjunctive support in the treatment of Graves’ disease, particularly during the recovery or remission phases.

**Figure 3 cimb-47-00696-f003:**
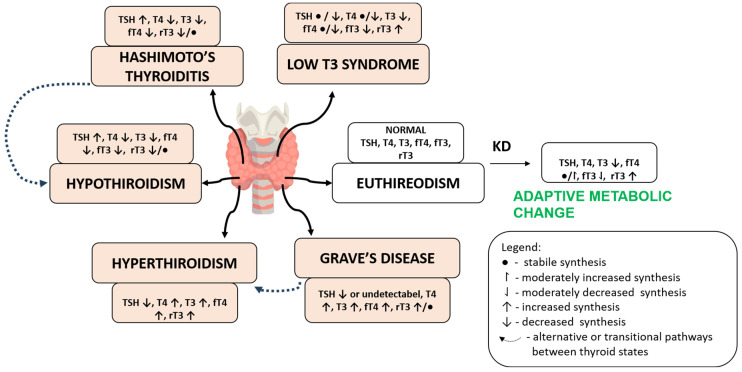
Biochemical parameters of thyroid disorders that could be affected by KD. KD in euthyroid individuals usually leads to an adaptive metabolic profile characterized by reduced T3 and fT3, elevated fT4, and stable or slightly reduced TSH. These adaptations differ from pathological thyroid disorders such as Hashimoto’s thyroiditis, hypothyroidism, or hyperthyroidism, in which the hormonal changes reflect a disease-specific dysfunction. Abbreviations: TSH—thyroid-stimulating hormone; T4—thyroxine; T3—triiodothyronine, rT3—reverse T3; fT3—free T3; fT4—free T4.

### 6.5. Thyroid Cancer

In the context of thyroid cancer, the KD has garnered attention as a potential adjunctive therapy due to its ability to exploit the metabolic vulnerabilities of malignant cells. Most cancer cells rely heavily on aerobic glycolysis—a phenomenon known as the Warburg effect—for rapid proliferation and survival, rendering them metabolically inflexible and highly dependent on glucose [122,123]. By restricting carbohydrate intake and inducing nutritional ketosis, the KD reduces glycolytic activity, lowers systemic glucose availability, and increases circulating ketone bodies, thereby creating a metabolic environment less favorable for tumor progression.

One of the most aggressive and lethal forms of thyroid cancer is anaplastic thyroid carcinoma (ATC), an undifferentiated T4-stage tumor known for rapid local invasion into structures such as the larynx, esophagus, carotid artery, and spine. Despite multimodal interventions (surgery, radiotherapy, chemotherapy), median overall survival is generally reported as 4–6 months, with only 10–20% of patients surviving past one year [124]. A preclinical study using a mouse model of ATC demonstrated that a ketogenic diet could suppress tumor growth, reduce tumor volume, lower lactate production, and increase apoptosis in malignant cells. These effects were particularly pronounced when KD was combined with the antioxidant N-acetylcysteine [125]. The rationale for these findings lies in the metabolic shift from glucose to ketone bodies: since cancer cells are metabolically reliant on glycolysis, ketosis deprives them of their primary energy source, potentially slowing their growth. This metabolic antagonism suggests a promising adjuvant role for the KD in the treatment of aggressive thyroid cancers. Additionally, ketone bodies BHB may inhibit HDACs, thereby exerting antiproliferative and pro-differentiation effects in thyroid cancer cells in vitro [126].

Despite these encouraging mechanistic and preclinical findings, the KD should be viewed strictly as an adjunctive, not curative, strategy. Human studies are currently lacking, and there is insufficient evidence to support the KD as a stand-alone treatment for thyroid malignancies. Furthermore, patients undergoing treatment for thyroid cancer—particularly those receiving systemic therapies, surgery, or radioactive iodine—have unique metabolic and nutritional demands. These patients require close monitoring to prevent unintended weight loss, sarcopenia, and micronutrient deficiencies. In select cases of refractory or rapidly progressing thyroid cancers, a KD may be cautiously considered alongside oncologic therapy under medical supervision. However, in well-differentiated thyroid cancers, such as papillary or follicular types with excellent prognoses, the clinical benefit of a KD remains speculative.

## 7. Risks, Monitoring, and Personalized Recommendations

Given the complex and individualized effects of KD on thyroid function, a system-level summary of the underlying mechanisms, biochemical mediators, and expected clinical outcomes is essential for translational utility (Table 1). The impact of KD spans multiple physiological domains including immune modulation [52,127], mitochondrial redox balance [128], hormone conversion pathways [129], nutrient sensing [130], and gut–liver–thyroid cross-talk [131]. While a KD may reduce inflammation, oxidative stress, and autoimmune reactivity in thyroid-vulnerable individuals, it also poses risks of impaired T4-to-T3 conversion, micronutrient depletion, menstrual cycle disruption, and misinterpretation of thyroid laboratory results due to functional central suppression. These effects vary according to genetics, sex, baseline metabolic status, energy availability, and duration or strictness of the diet. The table below integrates core pathways, relevant mediators, affected thyroid functions, and net clinical considerations.

Emerging clinical findings show how these mechanisms manifest themselves in practice. Controlled studies in healthy adults consistently show a reduction in T3 with a compensatory increase in T4, generally reflecting an adaptive rather than a pathological response [12,132]. In contrast, clinically relevant hypothyroidism has been reported in pediatric epilepsy cohorts with long-term KD, with a subset requiring levothyroxine therapy [108,118]. Some authors also suggest that the lower T3 during ketosis reflects improved metabolic efficiency rather than suppression [10]. Taken together, these findings highlight the importance of individualized monitoring, as described in the following recommendations.

The heterogeneous effects of KD on thyroid function call for a stratified monitoring approach that reflects each individual’s baseline endocrine status, genetic predisposition, and diet composition. Patients can be broadly categorized into three tiers of thyroid risk: (1) euthyroid individuals with no autoimmune markers, who generally tolerate a KD well with minimal monitoring beyond routine labs; (2) individuals with subclinical hypothyroidism, DIO2 polymorphisms, or borderline low fT3, who may require regular assessment of fT3, rT3, and relevant micronutrients; and (3) patients with overt hypothyroidism or autoimmune thyroiditis (e.g., Hashimoto’s disease), for whom a KD may be either beneficial or destabilizing depending on diet quality, energy balance, and adherence to repletion protocols. For the latter groups, baseline labs should include TSH, fT4, fT3, rT3, TPOAb and thyroglobulin antibodies, as well as selenium, zinc, iodine, magnesium, and ferritin. Follow-up testing should occur within 6–8 weeks of initiating KD, and again after any substantial change in caloric intake, macronutrient ratio, or weight loss. Clinical symptoms such as fatigue, cold intolerance, menstrual irregularities, or cognitive slowing should be taken as early indicators of tissue-level hypothyroidism even in the presence of normal serum TSH. In LT4-treated patients, particular vigilance is required: fT3 may drop despite unchanged TSH, especially in carriers of the *Thr92Ala DIO2* polymorphism that impairs T4-to-T3 conversion [133,134]. These individuals may benefit from combination therapy or dietary modifications that restore micronutrients and energy availability. Another clinical consideration is the rise in rT3 often seen during KD initiation or energy deficit, reflecting stress-driven conversion patterns that can blunt thyroid receptor activation. KD protocols should be individualized not only by macronutrient ratio but also by their inclusion of thyroid-supportive components, such as prebiotic fibers, unsaturated fats, organ meats, shellfish, and non-starchy vegetables. In certain high-risk groups, particularly lean women with low leptin or hypothalamic sensitivity, cyclical reintroduction of carbohydrates (e.g., therapeutic targeted carbohydrate intake every 7–14 days) may help maintain HPT axis resilience and prevent menstrual cycle disruption [135]. Ultimately, a personalized, feedback-driven monitoring strategy, integrating biochemical markers, clinical symptoms, and dietary variables, can help maximize the metabolic benefits of the KD while minimizing endocrine disruption.

## 8. Future Directions

The current evidence linking KD and thyroid function is constrained by several limitations. Most studies are of short duration, such as the three-week crossover study in healthy adults [12] or the 12-week modified Atkins intervention in drug-resistant epilepsy [42], and usually refer to small, specific populations. Pediatric epilepsy studies also report thyroid changes [108,118], but the results cannot be generalized to adults or non-epileptic cohorts. Even longer follow-ups, such as the 12-month study on GLUT1 deficiency and refractory epilepsy, remain limited to rare populations [15]. In addition, thyroid outcomes are often secondary endpoints, and marked heterogeneity between KD protocols and study groups makes comparisons difficult. These limitations highlight the need for adequately powered, long-term randomized trials specifically designed to evaluate thyroid outcomes in different KD variants.

Despite these limitations, the evolving understanding of KDs and their interaction with thyroid physiology presents both clinical opportunities and unanswered questions. In practice, KDs may serve as a useful adjunct in select populations, such as individuals with Hashimoto’s thyroiditis seeking autoimmune and metabolic stabilization, patients with obesity-related TSH elevation, or those with insulin resistance where improved leptin and glucose regulation may indirectly benefit thyroid function. In hypothyroid individuals on LT4 therapy, the KD may be appropriate but demands vigilant monitoring for fT3 suppression, particularly in carriers of DIO2 polymorphisms or those undergoing significant caloric restriction. Men and women with euthyroid status but distinct hormonal environments (e.g., postmenopausal women, athletes, or individuals with low body fat) may require individualized macronutrient structuring and micronutrient repletion to maintain thyroid homeostasis during prolonged ketosis. While early evidence from animal models suggests a role for KDs in inhibiting thyroid cancer progression via Warburg effect modulation, its translation into clinical oncology remains speculative and untested in large trials. Future research should prioritize controlled longitudinal studies exploring the effect of KD on thyroid hormone dynamics, autoimmunity progression, and patient-centered outcomes such as energy, cognition, and reproductive health. Genotype–phenotype interactions, especially involving DIO2, PPARA, and APOA2 variants, also warrant investigation to establish predictive markers for diet response. Additionally, mechanistic work is needed to clarify how SCFA levels, gut microbiota profiles, and bile acid signaling affect thyroid feedback under ketogenic conditions. From a clinical standpoint, the development of validated monitoring protocols that include rT3, fT3, and non-hormonal markers such as SHBG or leptin may improve safety and efficacy. Finally, tailored dietary templates—incorporating prebiotic fiber, therapeutic targeted carbohydrate intake, and micronutrient fortification—represent a promising avenue to retain the metabolic benefits of the KD while preserving thyroid and reproductive endocrine integrity.

## 9. Conclusions

The KD exerts multifaceted effects on thyroid physiology, acting at both central and peripheral levels through its influence on energy metabolism, immune regulation, and inflammatory tone. While reduced fT3 without elevated TSH appears to be a consistent adaptive response under carbohydrate restriction, this does not always reflect true euthyroidism, particularly in individuals with high stress loads, low energy availability, or genetic susceptibility. The KD may offer immunometabolic benefits in autoimmune thyroid conditions by reducing inflammation, oxidative stress, and improving regulatory T cell activity. However, it also carries potential risks, including functional hypothyroidism, micronutrient deficiencies, and hormonal imbalances, especially in lean, hormonally sensitive individuals or women. These bidirectional outcomes highlight the importance of personalized dietary strategies based on clinical presentation, comprehensive hormonal profiling, and nutritional adequacy. Future research should clarify the long-term safety and efficacy of the KD across different thyroid phenotypes, moving toward precision nutrition rather than universal recommendations.

## Figures and Tables

**Figure 1 cimb-47-00696-f001:**
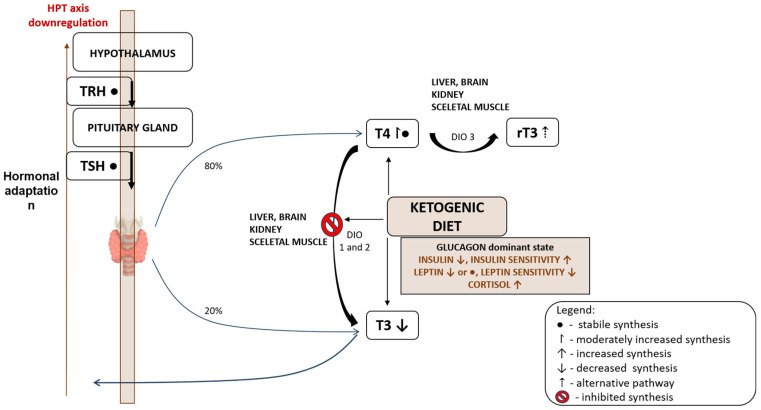
Overview of the mechanisms of interaction between the KD and the HPT axis. The KD reduces insulin and leptin signaling while transiently increasing cortisol, leading to downregulation of TRH and TSH secretion. In peripheral tissues, suppressed DIO1/2 activity reduces the conversion of T4 to T3, while increased DIO3 activity favors the rT3 formation. Abbreviations: HPT—hypothalamic–pituitary–thyroid; TRH—thyrotropin-releasing hormone; TSH—thyroid-stimulating hormone; T4—thyroxine; T3—triiodothyronine, rT3—reverse T3; DIO—deiodinase enzymes.

**Figure 2 cimb-47-00696-f002:**
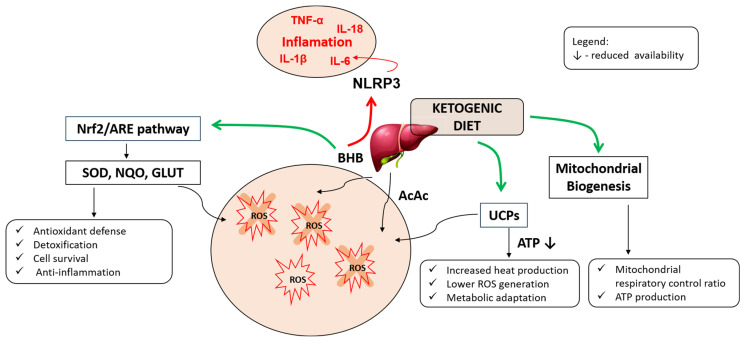
Effects of KD on oxidative stress and mitochondrial function. KD increases ketone bodies (BHB, AcAc), which reduce ROS formation, activates the Nrf2/ARE signaling pathway, and suppresses NLRP3 inflammasome activity, thereby reducing pro-inflammatory cytokines (TNF-α, IL-1β, IL-6, IL-18). In parallel, KD promotes mitochondrial biogenesis, upregulates uncoupling proteins and enhances metabolic efficiency, while reducing ATP production as part of adaptive energy conservation. Abbreviations: BHB—β-hydroxybutyrate; AcAc—acetoacetate; ROS—reactive oxygen species; Nrf2/ARE—nuclear factor erythroid 2-related factor 2/antioxidant response element; SOD—superoxide dismutase; NQO—NAD(P) H:quinone oxidoreductase; GLUT—glutathione transferase; NLRP3—NOD-like receptor protein 3 inflammasome; UCP—uncoupling protein; ATP—adenosine-triphosphate.

**Table 1 cimb-47-00696-t001:** Overview of KD effect on clinical implications related to thyroid function.

Mechanism/Pathway	Key Mediators	Thyroid Effect	Clinical Implication
Reduced glucose availability	↓ Insulin, ↓ Leptin	↓ DIO2 activity, ↓ T3	Lower peripheral T3, especially in LT4 monotherapy patients
Elevated ketone bodies	↑ BHB	↓ NLRP3, ↑ Tregs, HDAC inhibition	Reduced inflammation, possible autoimmunity modulation
Cortisol elevation (early KD)	↑ Cortisol	↑ rT3, ↓ T3 receptor activation	Functional hypothyroidism; fatigue, low body temperature
SCFA depletion (low fiber)	↓ Butyrate, Acetate, Propionate	↓ DIO2, ↑ permeability	Gut–thyroid axis disruption, impaired T4-to-T3 conversion
Bile acid dysregulation	Altered secondary bile acids	↓ Hepatic deiodinase activity	Reduced hormone clearance, disrupted liver feedback loop
Iodine and selenium intake variability	↓ Iodine, ↓ Selenium, ↓ Zinc	↓ T4 synthesis, ↓ antioxidant enzymes	Hypothyroidism risk, immune activation in Hashimoto’s
Leptin suppression	↓ Leptin	↓ TRH, ↓ TSH, ↓ GnRH	Central suppression; amenorrhea or thyroid downregulation
Energy conservation adaptation	↑ AMPK, ↑ SIRT1	↓ DIO1/2, ↑ mitochondrial efficiency	“Low T3 syndrome”; masked hypothyroidism despite normal TSH
Histone modification	HDAC inhibition by BHB	↑ Treg differentiation, ↓ IL-6/TNF-α	Immunomodulation; reduced thyroid antibody titers in some cases
Genetic predisposition	DIO2 Thr92Ala, PPARα, APOA2	Altered T3 conversion, lipid metabolism	Personalized response to KD; some genotypes requires T3 support

Legend: ↑ - increase, ↓ - decrease

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
