# Peer review of "Ketogenic Diet and Thyroid Function: A Delicate Metabolic Balancing Act"

_cimb, 2025, doi:10.3390/cimb47090696_

Round 1

Reviewer 1 Report

Comments and Suggestions for Authors

The manuscript provides a thorough and well-organized review of the intricate relationship between the ketogenic diet (KD) and thyroid function. The authors effectively synthesize current evidence on metabolic adaptations, hormonal regulation, immune modulation, and clinical implications, offering a balanced perspective on both the benefits and risks of KD for thyroid health. The review is timely and fills an important gap in the literature. However, several areas could be improved to enhance clarity, clinical applicability, and mechanistic depth.

  1. The review discusses various effects of KD on thyroid function (e.g., T3 suppression, immune modulation) but lacks specific, actionable guidance for clinicians.
  2. The role of KD in Graves’ disease (Section 6.4) is underdeveloped and somewhat speculative.
  3. The review does not distinguish between different types of ketogenic diets (e.g., classic KD, modified Atkins, cyclical KD), which may differentially impact thyroid function.
  4. Figures and tables could be more intuitive. For example, Figure 1 and Figure 2 use generic labels (e.g., "inhibition") without specifying key pathways.
  5. The review largely focuses on mechanisms and clinical observations but does not critically address limitations (e.g., reliance on small studies, lack of long-term data).

Author Response

Comment 1: The review discusses various effects of KD on thyroid function (e.g., T3 suppression, immune modulation) but lacks specific, actionable guidance for clinicians.

Response 1: Thank you for this comment. We would like to clarify that Section 7 of the manuscript already provides specific, actionable guidance for clinicians, including patient risk stratification, recommended laboratory markers and monitoring intervals, and dietary considerations. To make this even clearer, we have added a bridging paragraph after Table 1 in Section 7 that summarizes emerging clinical evidence and connects mechanistic pathways with practical monitoring strategies.

Comment 2: The role of KD in Graves’ disease (Section 6.4) is underdeveloped and somewhat speculative.

Response 2: We agree that the role of KD in Graves’ disease is largely theoretical and under-researched, and that there is a lack of substantial clinical evidence. We have therefore revised Section 6.4 to clearly emphasize this limitation.

Comment 3: The review does not distinguish between different types of ketogenic diets (e.g., classic KD, modified Atkins, cyclical KD), which may differentially impact thyroid function.

Comment 4: Figures and tables could be more intuitive. For example, Figure 1 and Figure 2 use generic labels (e.g., "inhibition") without specifying key pathways.

Comment 5: The review largely focuses on mechanisms and clinical observations but does not critically address limitations (e.g., reliance on small studies, lack of long-term data).

Response 5: Thank you for this important observation. We agree that the current evidence base is limited by small cohorts, short follow-up periods, and lack of long-term data. In the revised manuscript, we have added a critical appraisal of these limitations at the beginning of the Future Directions section. By integrating these examples, we underscore the need for long-term, adequately powered randomized trials specifically designed to investigate thyroid function under different KD variants.

Reviewer 2 Report

Comments and Suggestions for Authors

The manuscript by Vranjić et al. discusses the pros and cons of a ketogenic diet in thyroid hormone production. This manuscript is a well written, engaging article which I found highly interesting and easy to follow. I thoroughly enjoyed reading this manuscript. The review covers the role of hormones, diet, genetics, and the gut microbiome of TH regulation and it is thus complete and comprehensive. Diseases of the thyroid gland are also discussed in the context of a ketogenic diet. I have only minor comments to help with improving the final product.

Minor comments:

In figure 1, I suggest moving the words horizontally for easier reading

“Liver, Brain, Kidney etc…”

The review highlights the effects of leptin, insulin, cortisol and ghrelin on thyroid hormone, but it might be worth it to point out that there are others (in the 5th paragraph). While I agree that these ones are “major” regulators, other hormones (e.g. glucagon) also play a role in thyroid hormone stimulation and regulation).

Ln 233 is written awkwardly, perhaps reword this sentence.

Please define all acronyms in the figure captions – for example, figure 2 has many that could be defined.

In section 4.2, sex biases are addressed – but the focus is on hormones. Are there any genetic biases (or populations biases) that make individuals more or less susceptible to dietary modulation of thyroid hormone?

In section 6.4, the paragraph refers to hyperthyroidism but, in figure 3, it is spelled hyperthireodism. Is this correct?

Based on the statement in Ln 477 ("Gut microbial health is also critical for the absorption of thyroid-relevant micronutrients, particularly iodine (needed for T4/T3 synthesis), selenium (required for deiodinase and glutathione peroxidase) and zinc (involved in TSH production and receptor binding)", do we know which bacteria are beneficial for these activities? If probiotics were used, could they improve thyroid hormone production or mitiagte adverse effects of a Ketogenic diet?

Author Response

Comment 1: In figure 1, I suggest moving the words horizontally for easier reading “Liver, Brain, Kidney etc…”

Response 1: Thank you for this comment. The words in the Figure have been placed horizontally.   

Comment 2: The review highlights the effects of leptin, insulin, cortisol and ghrelin on thyroid hormone, but it might be worth it to point out that there are others (in the 5th paragraph). While I agree that these ones are “major” regulators, other hormones (e.g. glucagon) also play a role in thyroid hormone stimulation and regulation).

Comment 3: Ln 233 is written awkwardly, perhaps reword this sentence.

Response 3: Thank you for this suggestion. The sentence has been rephrased.

Comment 4: Please define all acronyms in the figure captions – for example, figure 2 has many that could be defined.

Response 4: Thank you for pointing this out. All acronyms have been defined in the figure captions to ensure clarity of the figures.

Comment 5: In section 4.2, sex biases are addressed – but the focus is on hormones. Are there any genetic biases (or populations biases) that make individuals more or less susceptible to dietary modulation of thyroid hormone?

Response 5: Genetic determinants of thyroid adaptation to KD, including DIO2, APOA2, and PPARA polymorphisms, are addressed in Section 4.1. To complement this, Section 4.2 has been expanded to highlight sex-specific and population-level differences.

Comment 6: In section 6.4, the paragraph refers to hyperthyroidism but, in figure 3, it is spelled hyperthireodism. Is this correct?

Response 6: Thank you for noticing this typographical error. The term in Figure 3 has been corrected from “hyperthireodism” to “hyperthyroidism” to ensure consistency with the text.

Comment 7: Based on the statement in Ln 477 ("Gut microbial health is also critical for the absorption of thyroid-relevant micronutrients, particularly iodine (needed for T4/T3 synthesis), selenium (required for deiodinase and glutathione peroxidase) and zinc (involved in TSH production and receptor binding)", do we know which bacteria are beneficial for these activities? If probiotics were used, could they improve thyroid hormone production or mitiagte adverse effects of a Ketogenic diet?

Response 7: We have clarified this point in Section 5. While gut microbial health is indeed critical for iodine, selenium, and zinc absorption, specific bacterial taxa directly supporting these processes in humans remain insufficiently characterized. Preliminary data suggest that probiotics, particularly Lactobacillus and Bifidobacterium strains, may improve micronutrient handling and reduce thyroid autoantibodies, but these findings are still limited and have not been tested in the context of KD. We have clarified this point in Section 5.